# A Comprehensive Superposition of Viral Polymerase Structures

**DOI:** 10.3390/v11080745

**Published:** 2019-08-13

**Authors:** Olve B. Peersen

**Affiliations:** Department of Biochemistry & Molecular Biology, Colorado State University, Fort Collins, CO 80523-1870, USA; Olve.Peersen@ColoState.edu; Tel.: +1-970-491-0433

**Keywords:** polymerase, structure, structure alignment

## Abstract

Nucleic acid polymerases are essential enzymes that replicate the genomes of both RNA and DNA viruses. These enzymes are generally encoded by viruses themselves so as to provide biochemical functions and control elements that differ from those of the host cell polymerases. The core active site structure used by all replicative polymerases is highly conserved and composed of two key aspartate residues from the conserved motifs A and C, but beyond this there is significant divergence among structures. These differences can make it difficult to select which portions of structures to align for comparisons, yet there are extended structural similarities within different groups of viral polymerases that should clearly be considered to generate optimal alignments. This manuscript describes a comprehensive structure-based superposition of every viral polymerase structure solved thus far based on an alignment-tree approach wherein aligned regions grow in complexity as similarity among polymerases increases. The result is a set of 646 structures that have been aligned into a single common orientation. This provides a convenient resource for directly comparing viral polymerases and illustrating structural conservation among them. It also sets the stage for detailed bioinformatics analysis to further assess common structural features. The full set of protein data bank (PDB) formatted files is publicly available via the Polymerase Structures community page at the Zenodo.org open data repository.

## 1. Introduction

Nucleic acid polymerases play a central role in the replication of all viruses and bacteriophages. These essential enzymes exist in forms that synthesize DNA or RNA using DNA or RNA templates, resulting in their classifications as RdRP, i.e., RNA-dependent RNA polymerases, and analogously RdDP, DdRP, and DdDP enzymes. Viral polymerases are generally built on the same structural framework as the replicative host cell polymerases, retaining the palm, thumb, and finger domains named by analogy to the structure of a human right hand (Figure 1). These structural domains in turn contain the hallmark conserved motifs A–G that readily identify polymerases based on protein sequence, and also allows for their structural superpositioning via the highly conserved core active site region. This article does not extensively reference individual structures and/or conclusions made from them, and does not discuss the functions of the conserved motifs in great detail, but the accompanying review by Selisko et al. [1] in this issue of *Viruses* nicely describes this in the context of flaviviral RdRPs.

Rather, this article describes the method underlying a comprehensive superpositioning of viral polymerases yielding a set of 646 structures from 414 different protein data bank (PDB) entries released up until 30 June 2019. Within this collection, any one structure can easily be compared with any other by simply loading their PDB files into a molecular graphics program. The collection includes superpositions of closely related polymerases that allow for easy visualization of differences among them, and superpositions of multiple structures of the same protein to easily assess changes in structure arising from mutations, differences in crystallization conditions, or bound ligands.

The complete set of superimposed polymerase coordinates are publicly available in the community site Polymerase Structures at Zenodo, the Open Science platform at the CERN Data Centre (http://www.zenodo.org/communities/pols/) [2]. These alignments will be regularly updated as new structures become available and/or as readers offer suggestions for new or improved comparisons. Readers can also submit their own related analyses of structures via this community site. In addition, many of the figures in this article are accessible in a dynamic and interactive form via a WebPyMol interface, as listed in the Appendix A. The PyMol session files used for these figures are also downloadable, allowing readers to manipulate them on their own computers using the full or educational versions of the PyMol program available via www.pymol.org. 

## 2. Polymerase Structure Overview

The palm domain forms the polymerase core and contains the three stranded anti-parallel β-sheet with two key aspartate residues that make up the catalytic center of the enzyme. Together with the incoming NTP, these aspartates coordinate two Mg^2+^ ions that help stabilize the deprotonated form of the priming 3’ hydroxyl group, facilitating its attack on the NTP α-phosphate that leads to elongation of the product strand by one nucleotide. This polymerase mechanism and associated active site geometry are highly conserved across replicative polymerases [3], and are maintained in essentially all viral polymerases. The fingers domain plays multiple functional roles: a) it binds the downstream nucleic acid, guiding the template strand into the active site, b) it positions the templating nucleotide for base pairing with the incoming NTP substrate, c) in many polymerases the fingers undergo a large molecular motion that repositions the templating base–NTP base pair over the active site for catalysis, and d) the fingers provide an important basic lysine or arginine residue that interacts with the NTP α-phosphate group from above the active site to help position the NTP in the active site for catalysis. The thumb domain forms the primary interface for interactions with the upstream product nucleic acid, and an α-helix from the thumb often lies in the minor groove of the exiting duplex for viral polymerases that do not have a strand separation mechanism.

## 3. Structure Superposition Tree

There are two fundamental issues for generating reliable structure superpositions; first, identifying the residues that should be considered structurally equivalent, and second, reorienting all the individual structures to optimally align these matched residues with each other. While it is straightforward to visually appreciate that viral polymerases share the core conserved active site motifs, it is somewhat more complicated to develop a comprehensive superpositioning scheme that would allow comparisons across all the structures solved to date. One could pick a “master” polymerase and pairwise align all other polymerases to this, but such a comparison would be limited to a rather small region of strong structural similarity. A better approach would be to use those parts of the entire structures that share the same fold. This means the superposed regions could become larger when the structural similarity is greater, and in the extreme case of comparing many structures of the same polymerase it would be best to use complete structures. For the more common case of somewhat divergent polymerases it would be best to use only their structurally conserved regions, for example the core motifs A, B, and C that surround the active site. However, a major downside of this approach is that it requires a meticulous one-to-one mapping of structurally equivalent residues, which becomes more difficult as structures diverge. In fact, the diversity among all viral polymerases means that only a small portion of their structures within motifs A and C could be considered truly similar for such a global comparison, as illustrated by Figure 2 and Appendix A.

Consideration of these factors led to the development of a tree-based organization for polymerase structure alignments wherein each node on the tree is a multiple structure alignment that inherits its orientation from the previous node on the tree. The overall tree is illustrated in Figure 3, where the nodes have four-letter names corresponding to alignment “**sets**” that will be indicated in boldface throughout the text. The tree starts with the **pols** set that is a superpositioning of minimal shared core segments from the four major classes of DNA- or RNA-dependent DNA or RNA polymerases, i.e., DdDP, DdRP, RdDP, and RdRP enzymes. Two different DdDP structures were included, one from an A family polymerase (DDDA, T7 DNA polymerase) and one from a B family polymerase (DDDB, RB69 polymerase). The orientations of the five structures resulting from this alignment are then retained as each of them seeds one of the next sets that superpose more closely related polymerases. These have larger regions of similarity, which will expand and improve the subsequent alignments. For example, the RdDP structure from the **pols** set becomes the master orientation for the subsequent **rddp** set that superposes reverse transcriptase structures from HIV, Moloney murine leukemia virus (**mmlv**), and telomerase. This process is repeated as necessary to build out the tree, with the branches ending in superpositions of either very similar proteins, such as the **flav** set with all flaviviral polymerase structures, or multiple structures of the same protein, such as the **ev71** set with all available enterovirus 71 polymerase structures. Each superposition set named in Figure 3 corresponds to a single multiple structure alignment, and comparisons of structures within a single set will be the most reliable. But the consistent mapping back to the parental orientation in the tree means that one can mix and match PDB coordinate files from distantly related branches and retain fairly reliable alignment. The overall result is 646 different coordinate files corresponding to 646 different polymerase chains taken from 414 different original PDB files, all of which have been reoriented into a common coordinate space.

### 3.1. Maximum Likelihood Superpositions with Theseus

The second major issue is the mathematical treatment used for the alignment itself, i.e., how does one determine the reorientation matrix that is used to move a structure from its original orientation into a new orientation where it is superimposed on another structure. For this I chose to use maximum likelihood based multiple structure alignments with the program Theseus [4,5,6]. Unlike a traditional pairwise least-squares alignment of two structures where every atom position is weighted equally, the maximum likelihood method simultaneously aligns multiple structures and considers the statistical distributions of coordinates at every position to arrive at a globally optimal fit. Without going into the mathematical details, which are well described in [7], the end result is to effectively down weight structurally divergent regions and converge about the most structurally similar regions. This effect can be appreciated by the superposition comparison figure shown in [5] and on the www.theseus3d.org web site, where a family of NMR structures is used to show that the maximum likelihood (ML) alignment is clearly superior to the classical least-squares (LS) alignment that is biased by the conformational heterogeneity in internal loops and the ends of the protein.

For the polymerase superposition tree, the branch termini are usually groups of identical or closely related polymerases with tight distributions of conformations because the structures are fairly similar, e.g., **flav** set with NS5 structures from dengue, West Nile, zika, and Japanese encephalitis viruses as shown in Figure 4. Within such groups the similarities in both sequence and structure space are high enough that the full-structure superpositioning process can be automated by the method outlined in Section 4. This basically consists of assigning equivalent residues via a multiple sequence alignment from Muscle [8] and then using Theseus to align the structures in three dimensions based on Cα positions only. Notably, while minor errors in the sequence alignment can result in mis-assigning structurally equivalent residues, this does not significantly affect the final superposition because such positions will mathematically appear as divergent parts of the structure and consequently be effectively down-weighted in the analysis.

### 3.2. Picornaviral Polymerases as an Example

Figure 5, Figure 6, and Appendix A further illustrate the structure superposition hierarchy starting with the **pols** set and ending with **poli** that contains all solved poliovirus polymerase structures. Whereas Figure 4 showed the individual structures for each step in the hierarchy, Figure 5 shows the average structure as a cartoon “worm” with spheres for the Cα atoms of the two conserved active site aspartate residues in motifs A and C. To give a visual indication of structural heterogeneity across the superposed region, the worm is colored according to a pseudo B-factor calculated as (8π^2^) times the statistical variance observed at each Cα position. The representative structures chosen for each set contained a canonical conformation across the superposed region so as not to bias the resulting orientations.

The first stage superposition set (**pols**) provides the base orientations for representative structures of the four classes of polymerases, and it is done using minimal 13-residue versions of the core motifs A and C that represent the limited extent of active site structural similarity among these highly divergent polymerases. The RdRP structure from this superposition is then used to anchor the orientation of the **rdrp** set consisting of three different viral RdRPs—one each from the positive strand (**psrn**), negative strand (**nsrn**), and double stranded RNA (**dsrn**) viruses. This **rdrp** set expands motifs A and C from 13 to 26 residues and adds 14 residues from motif F, the pair of antiparallel β-strands found above the active site in RdRPs, resulting in a total of 40 residues (also shown in Figure 4). The subsequent **nsrn** set was expanded with an influenza virus specific set and the **dsrn** set was further expanded with five sets corresponding to different dsRNA viruses (see Figure 2). Bacteriophage Qβ was given its own **qbta** set by superimposing all Qβ replicase structures onto core motifs A, C, and F from the representative RdRP structure used in the **pols** set.

At this point there is a significant divergence in the active site structures of the positive strand RNA virus (**psrn**) polymerases compared to all the other groups of polymerases. This arises because these enzymes close their active sites for catalysis by a subtle movement of motif A that completes the β-sheet between motifs A and C [9]. The default structures of most **psrn** RdRPs, solved in the absence of RNA and NTPs, reflect the open state of the active site. However, several of these RdRPs have been captured in the closed state when crystallized as elongation complexes with RNA and a correct nucleotide triphosphate. While the differences between the two states are structurally subtle, as described in Section 5.6, they are significant at the level of active site superpositions, particularly in the initial **pols** and **rdrp** sets where the core active site structures of other polymerase classes reflect the closed state of a **psrn** RdRP. To accommodate this structural transition in the alignment tree, a special superposition set called **opcl** was devised to interconvert between the open conformation that is predominant among the positive strand virus RdRP structures and the closed conformation found in the prior alignment sets (**pols** and **rdrp**) of the tree. To limit bias from any one pair of open versus closed structures, this **opcl** set was assembled from eight different structures that represent two pairs each of poliovirus and enterovirus 71 3D^pol^ structures that were solved in both the absence and presence of NTP by soaking experiments of identical crystals [9,10]. The complete structures were used such that the conformational changes associated with the open vs. closed active sites did not significantly affect the alignment. The orientation inherited from the **rdrp** set is a closed form EV71 polymerase structure (PDB: 5F8J) and the corresponding open form structure (PDB: 5F8G) is used as the seed orientation for the subsequent alignment of all positive-strand RNA polymerases. 

The next **psrn** set is a superposition of six different positive-strand RNA virus polymerases and the aligned region has been expanded from 40 to 65 residues by including motif B that forms a loop structure followed by a long α-helix that lies adjacent to motifs A and C. The representative structures in this **psrn** alignment in turn define the parent orientations for subsequent alignments of caliciviral (**cali**), coronaviral (**coro**), flaviviral (**flav**), hepatitis C (**hepc**), picornaviral (**pico**), and *Thosea asigna* virus (**tavp**) polymerases. A set for the noroviruses (**noro**) was generated under the **cali** set, and a pestivirus set (**pest**) was made under the **flav** set. The **pest** set used the representative flaviviral structure (FLAV.pdb) from **psrn** as an alignment seed to avoid over-representing flaviviral polymerases in the **psrn** set. Note that *Thosea asigna* virus polymerase has an altered topology where motif C precedes motif A in the primary sequence, but the active site conformation is conserved nonetheless. There are interesting structural parallels between this and the double stranded RNA birnavirus (**birv**) polymerase that shares this non-canonical topology [11].

Finally, the picornaviral polymerases are aligned in two stages that end in sets superposing all available structures of each viral polymerase (Figure 6). First, the **pico** set is composed of three structures representing encephalomyocarditis virus (**emcv**), foot-and-mouth disease virus (**fmdv**), and poliovirus 3D^pol^ as a representative enterovirus (**entv**). This was done because there is a difference in the orientation of a pinky finger helix for these structures, with FMDV and EMCV being similar to each other yet distinct from the enteroviruses (Appendix A, scene F5). The second stage is an **entv** set with six polymerases that give rise to separate coxsackievirus B (**coxb**), enterovirus 71 (**ev71**), poliovirus (**poli**), and rhinovirus (**rhin**) alignment sets. Each of these enterovirus sets in turn superimpose all available structures of each polymerase. The picornaviral polymerase alignments are done using the complete ≈460-residue structures, not just the core motifs as for previous sets in the tree, and they are based on a manually curated structure-based sequence alignment (the file 4-pico/sav/pico_v5.aln, Appendix A). This alignment assigns all structurally equivalent residues and accounts for both residue insertions/deletions and regions with significantly different conformations, e.g., the pinky finger helix mentioned above, by making non-overlapping sections in the sequence alignment so that the structure alignment does not attempt to superpose these divergent structures on each other. The single available EV-D68 structure (PDB: 5XE0) was included in the **entv** set, and the single available coxsackievirus A16 structure (PDB: 5Y6Z) was included in the **ev71** set. 

### 3.3. Small Molecule Inhibitor Complexes

The superposition sets for the viral polymerases are fairly comprehensive as of the time of writing, with the exception of the hepatitis C polymerase (**hepc**) and HIV-1 reverse transcriptase (**hiv1**) sets that are composed of several representative structures. For these two proteins there are an exceedingly large number of structures available due to their pharmaceutical importance and associated antiviral compound development, and it is impractical to include all available structures. However, based on the classification of hepatitis C polymerase inhibitors by Venkataraman et al. [12] in this issue of *Viruses*, four additional sets that reflect classes of inhibitor-bound structures were generated. These are kept distinct from the **hepc/** directory by placing them in a **hepc_inhibitors/** directory, where they are divided into inhibitors binding to the palm (**hcip**), thumb (**hcit**), primer grip (**hcig**), or interfaces (**hcii**). Note that these additional 206 inhibitor complex structures arising from 109 distinct PDB entries are not included in the _All_PDBs/ directory described below.

## 4. Alignment Procedure and Output File Structure

The semi-automated procedure used to generate the superposed structures is diagrammed in Figure 7, and more detailed information about the resulting file and directory structures are given below. In brief, alpha-carbon (Cα) coordinates for every polymerase chain in a given PDB are extracted to separate files, a multiple sequence alignment is generated using the program Muscle [8], the structures are superposed with the program Theseus using only the Cα atoms with residue-equivalence mappings taken from the sequence alignment, and the new orientations of the structures are saved as a set of Cα-only PDB files. PyMol is then used to superpose the complete original PDB entry onto the rotated Cα structure, resulting in an output file that contains the full content of the original PDB file, but now reoriented according to the polymerase chain. If there are multiple polymerases in the original PDB entry then there will be one alignment for each polymerase chain, i.e., there will be multiple aligned structure files arising from a single PDB entry. No check is made for structures solved using strict non-crystallographic symmetry; if that is the case then several of the resulting chains will have the exact same structure.

All the reoriented coordinates, as well as several associated output files, are publicly available via the Polymerase Structures community page (http://www.zenodo.org/communities/pols/). These will be regularly update as new structures are solved and/or readers point out omissions or errors on my part. The overall directory structure and file content are described below, and efforts have been taken to make it moderately self-explanatory, transparent, and consistent, hopefully making it a suitable resource for further analysis by other researchers. 

For users who simply want to obtain all the superposed coordinates, the _All_PDBs/ directory contains all the PDB files in one place, logically named using the original PDB codes with added terms to indicate the protein chain and superposition set used to generate the file. For the more advanced reader there is a collection of directories that contain all the files pertaining to each individual superposition set (Figure 8 and Figure 9). These include the original PDB entries, the sequence alignments, the final superposed coordinate files, and a number of intermediate files that may be useful for further analysis. These sets have logical four-letter names such as **pico** for picornaviral polymerases and **rb69** for all the bacteriophage RB69 polymerase structures, which are also the set names shown in Figure 2. Note that the directory names for the early superposition sets, i.e., those containing only the core substructures from divergent species, are preceded by a number, e.g., **1-pols**, **2-rdrp**, **3-dsrn**, to both indicate their level within the alignment tree and to sort them hierarchically at the top of the directory listing. These numbered sets are also colored red in Figure 2, where their leading numbers were omitted for clarity.

Each superposition set directory contains a **_readme.txt** file that lists the TITLE records of all the PDB files within that set as well as the residue ranges and number of atoms used for the superpositioning. All REMARKs and other header information except the TITLE flag have been stripped from the output files and the original symmetry operators should not be applied as they are no longer valid. The PDB entry portion of the filenames are in lower case for normal Protein Data Bank entries, and in upper case for the representative structures that are used to seed orientations along the alignment tree. For example, the **2-rdrp** set contains DSRN.pdb, NSRN.pdb, and PSRN.pdb for representative double, negative, and positive strand RNA virus polymerases, respectively, in addition to the RDRP.pdb file that was inherited from the previous **1-pols** set. These representative structure files contain a TITLE line that includes the PDB code of the entry from which the coordinates were extracted.

The aligned coordinate files are named using “PDB_Chain-**Set**.pdb” convention and contain the complete content of the original PDB entry in the new orientation. For example, the PDB entry 5D98 contains two copies of the influenza C virus PB1/PB2/PA polymerase complex, giving rise to two output files within the influenza virus **fluv** alignment set, one being superposed by the B chain PB1 molecule and the other by the E chain PB1. These two files are located in a pdb/ subdirectory of the fluv/ directory and called 5d98_B-fluv.pdb and 5d98_E-fluv.pdb.

### Superposition Directory Contents

Each superposition set directory has a standard composition of files and directories, as shown in Figure 9 for the **cypo** set. The pdb/ subdirectory within each set contains all the output structures and provides for easy “drag-and-drop” comparisons in a molecular graphics program. In addition, the files in the pdb/ subdirectories of all the sets have been duplicated into the _All_PDBs/ directory located at the top level, where they sort by PDB entry based on the initial portion of the filenames while the ends of the filenames indicate which superposition they come from.

## 5. Example Analyses of Superposed Structures

Using the set of superposed structures, one can further analyze them pairwise for direct comparisons, look at conservation of certain interactions or conformations across multiple related structures, and do very large scale analyses of all the files in the _All_PDBs/ directory. This can be done manually with suitable graphics software, or via automated scripting to parse the files for specific atoms or interactions based on spatial positions. The examples below illustrate some of the ways this can be achieved.

### 5.1. Specific Structure Comparisons

A primary application of the superposition set is to quickly compare structures, which could be highly divergent polymerases from multiple sets, or very similar structures from within the same set. One advantage of the superposed coordinates is that all the structures will be viewed in the same orientations, facilitating direct comparisons among them. For example, interactive Appendix A shows all the picornaviral polymerases, highlighting both their strong structural similarity and the differences in the pinky finger helix of the enterovirus versus apthovirus enzymes. Figure 10 shows a comparison of the picornaviral poliovirus 3D^pol^ elongation complex and the flaviviral polymerases that highlights how the “priming loop” from the flaviviral thumb domain occupies the duplex RNA exit channel as observed in 3D^pol^. It is this loop that prevents the use of duplex primer-template RNAs as primers for in vitro elongation reactions.

### 5.2. Sequence Assisted Structure Alignments

Another use is interpreting sequence alignments in a structural context to easily identify if residue differences are located on the protein surface, in the interior, near the active site, near RNA binding interfaces, etc. Insertions/deletions are commonly located at surface exposed loops that tend to be structurally flexible, and an examination of the sequence in the context of a collection of structures can be used to more accurately assign equivalent residues in the sequence alignment. Such an analysis was used to derive the customized sequence alignment of the picornaviruses, as described above for the **pico** set to generate the 4-pico/sav/pico_v5.aln alignment file (Appendix A).

### 5.3. Structural Heterogenity and Implied Flexibility

A secondary application is to easily assess if specific residues are located in well-ordered or flexible regions of the structures. Due to their size, most of the polymerase structures have been solved by X-ray crystallography, a technique that by its very nature cannot provide much information about structural flexibility. There certainly are cases where strong electron density can be used to place some residues in specific “alternate conformations,” but in general protein flexibility will result in localized weak to non-existent electron density because the molecules in the crystal itself do not have the same atomic positions. The conformations of residues modeled into such weak density will usually differ among structures, and the resulting structural heterogeneity reflects the inherent structural flexibility of these regions. In the polymerase superposition sets this can be visualized in two different ways; 1) by overlaying a large set of structure as backbone ribbons to directly observe the different conformations, or 2) by loading the average structure (pdb/theseus_ave.pdb) and coloring by its B-factor field that reflects (8π^2^)(variance) of each position in the Theseus superposition. Both approaches are shown in Figure 4, Figure 5 and Figure 6 and the variances are also listed in the file super/theseus_variances.txt associated with each superposition set.

### 5.4. Region Specific Selection of Atoms

A major advantage of having all the polymerases in essentially the same orientations is that one can make atom selections based on regions in real space that will be universally located on all the structures. For example, the Cα atom of the first aspartate residue in the active site GDD motif is located at (x, y, z) coordinates of approximately (10.0, −16.3, −13.8), while the Cα of the motif A aspartate is at approximately (3.5, –17.0, –11.5). Similarly, the center of a base pair between an NTP in a **psrn** active site and the templating nucleotide is located at approximately (14.5, −4.5, −15.5). Within PyMol, one can create pseudoatoms at these positions and then use them as anchor points for regioselection of more specific parts of the structures or for distance measurements. For example, the following set of commands will effectively trim the superposed structures down to show only the one polymerase chain used for the superposition and any bound nucleic acid or other chains:

pseudoatom pseudoBP, pos = [14.5, −4.5, −15.5]

show sphere, PseudoBP

color yellow, PseudoBP

select Monomers, PseudoBP expand 22

select Monomers, bychain Monomers

hide everything, not Monomers 

remove (not Monomers)     [optional command to delete atoms not in “Monomers”]

This set of commands creates a new pseudoatom (PseudoBP) in the middle of the nascent base pair in the active site, shows it as a yellow sphere, creates an object “Monomers” composed of all chains that have at least one atom located within 22 Å of this pseudoatom, and then hides the display of all atoms that are not part of the Monomers object. The optional last command can be used to delete the non-selected atoms from the PyMol session altogether, i.e., remove all the additional molecules found in the structure files. Note that these commands work for most structures, but there may be some cases where the “expand 22” Å cutoff must be reduced to avoid inadvertently selecting nearby polymerase chains, or perhaps increased to select additional nearby chains.

### 5.5. Global Analysis of Conformations

The palm domain based active site closure step that is unique to the positive strand RNA viruses presents a good example of how a global analysis of the entire polymerase structure collection can be used to analyze active site conformations via distance measurements. The first two PyMol commands below will create a pair of pseudoatoms at the approximate locations of the Cα atoms of the two key active site aspartate residues located in motifs A and C. The next two commands select the true aspartate residue Cα atoms in all currently open structures by using the “expand” option to limit the search to atoms that are within 2.0 or 2.5 Å of each pseudoatom; the larger 2.5 Å limit is used for the motif A aspartate to accommodate its movement between the open and closed active site conformations of the positive strand RNA virus RdRPs, as previously described for the **opcl** set. 

pseudoatom PolAspA, pos = [3.5, −17.0, -, −11.5]

pseudoatom PolAspC, pos = [10.0, −16.3, −13.8]

select AspA, PolAspA expand 2.5 and resn asp and name ca

select AspC, PolAspC expand 2.0 and resn asp and name ca

Further scripting in PyMol can then be used to extract the exact residue numbers of each aspartate, and using those numbers one can compose command arguments to measure the distance between them. Combined with a script that sequentially loads all the PDB files in the All_PDBs/ directory, this can be used to calculate specific inter-atomic distances in all the polymerase structures and then carry out statistical analyses to examine differences in active site conformations, as described below.

### 5.6. Positive Strand RNA Virus RdRP Active Site Closure via Motif a Movement

Figure 11 shows an example of such an analysis to examine the distances between the active site motifs A and C in all polymerases, and then breaking these down further by specific classes or virus families. These two classic polymerase motifs form anti-parallel β-strands and their relative conformations can be assessed via Cα•••Cα distances along the protein backbone, as illustrated by 11A, or alternatively by a single distance “DD” between the Cα atoms of the motif A and C aspartates that coordinate the two Mg^2+^ ions during catalysis. Overall, a broad range of DD distances ranging from 5.3 to 8.4 Å is observed, as shown by the histogram and dot plot representations in Figure 11B.

The above analysis of distances between the motif A and C aspartates make it clear how unique the open conformation active site is to the positive strand RNA virus RdRPs as compared to all other viral polymerase, whether they be RNA or DNA based. The default structure for these enzymes is an open state active site where the anti-parallel β-sheet hydrogen bonding patterns between motifs A and C is not fully established and the motif A aspartate sidechain is pointed away from the active site. At this point several picornaviral structures have been solved in the closed state where the distances are equivalent to those found in the canonical polymerases. 

### 5.7. Active Site Metal Ions

The collection of superposed structures can also be used to analyze ions bound to commonly occupied sites. Figure 12A and Appendix A show views of the poliovirus RdRP active site with all the metal ions extracted from all the PDB files in the superposition set. This is done via a Unix grep search of the files in All_PDBs/ and includes a range of ion valencies and charges; the individual coordinate lines in the resulting IONS-**set**.pdb files have been appended with the names of the source PDB files to make it easier to identify their original structures.

The clustering of these ions shows there are three major metal binding sites near the active site (Figure 12B); two of these correspond to the classic metal A (meA) that is delivered in a complex with NTP and metal B (meB) that is pre-bound to the polymerase [3]. The third position (meB’) is offset from the active site and primarily found in viral RdRPs, as shown by the further breakdown of the bound metals by polymerase type in Figure 12C. 

Based on picornaviral polymerase structures solved with open and closed active sites, this third position likely represents a non-catalytic site for metal B that is either an initial binding site or an open conformation storage site. The ion would then move into the active site during active site closure via electrostatic interactions with the motif A aspartate so that is can function as metal B during catalysis. The movement of this aspartate during active site closure is shown in the **psrn** panel of Figure 12C, where the aspartate is pointed toward the meB site in the closed (yellow) conformation, but the alternate meB’ site in the open (green) conformation.

## 6. Final Comments

This comprehensive structure alignment represents the publication of an ongoing project aimed at providing an easy to use platform for comparing viral polymerase structures with coordinate sets that are readily accessible to even novice users. The collection of coordinate files will be updated in the future as more structures are solved, and may be expanded with non-viral polymerase branches. Indeed, the basic tree methodology used to align the polymerase structures provides a framework for similar superpositions of other classes of proteins where there is significant structural homology within a core region, but then highly divergent structures beyond this core. 

This article has provided a few examples of analyses that can be carried out, such as active site distance measurements across all polymerases or pairwise structure comparisons. Members of the scientific community are invited to carry out their own analyses of the structures, and if desired these can be uploaded to the Polymerase Structures community site and given a permanent digital object identifier (DOI) record by Zenodo.org as an open access submission of a dataset.

For ease of downloading and use, the superposed structures are available as either a single compressed archive of just the _All_pdbs_v2/ directory with all the reoriented PDB files (except the HCV inhibitor complexes), or a compressed archive of the entire collection of directories and files. The initial public release now available on the Polymerase Structures community page is listed as Version 2 (_v2) and minor changes and error corrections to it will be labeled as v2a, v2b, etc. Major changes at some point in the future will be designated as _v3 and beyond.

## Figures and Tables

**Figure 1 viruses-11-00745-f001:**
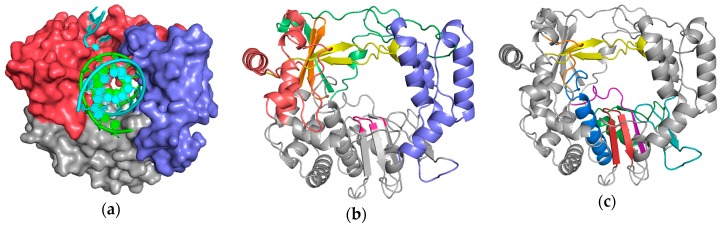
Poliovirus 3D^pol^ as a representative viral polymerase structure in complex with nucleic acid: (**a**) the major structural domains known as the palm (grey), fingers (red), and thumb (blue) are named by analogy to the shape of a cupped right hand; (**b**) identification of the individual finger structures as index (green), middle (orange), ring (yellow), and pinky (red), and the active site YGDD motif in the palm is colored magenta. (**c**) Conserved sequence motifs A–G mapped onto this structure. The palm domain core is composed of motifs A (purple) and C (red) that form a 3-stranded β-sheet folded up against the motif B (blue) α-helix. Motif D (green) forms the outer rim of the NTP entry channel, motif E (teal) is at the junction of the palm and thumb and also known as the “primer grip”, motif F (yellow) lies above the active site, and motif G (orange) lines the RNA entry channel. Appendix A allows you to dynamically rotate these structures in real time via a web-based version of the molecular graphics program PyMol.

**Figure 2 viruses-11-00745-f002:**
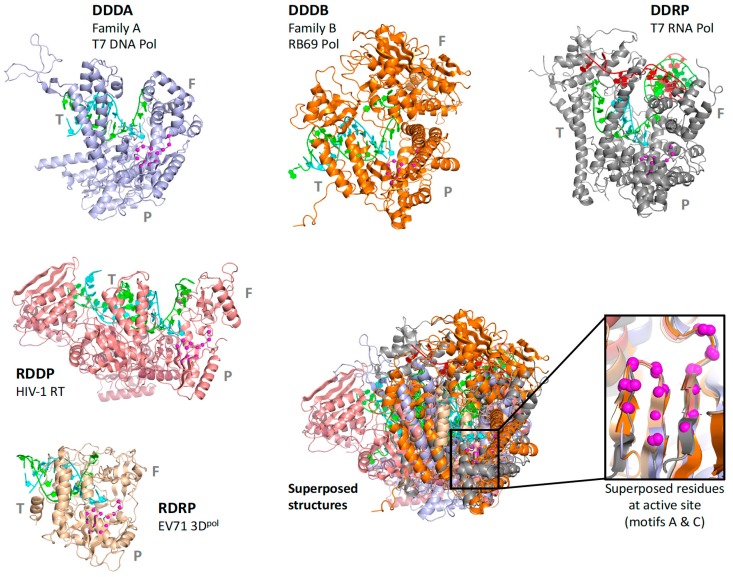
Diversity of polymerase structures demonstrated by five viral polymerases representing DNA-dependent DNA polymerases from family A (DDDA) and family B (DDDB), a DNA-dependent RNA polymerase (DDRP), a RNA-dependent DNA polymerase (RDDP), and a RNA-dependent RNA polymerase (RDRP). The structures are drawn to scale, P/T/F indicate palm/thumb/fingers domains, and the inset shows magenta spheres for the Cα atoms in motifs A and C used for the superposition.

**Figure 3 viruses-11-00745-f003:**
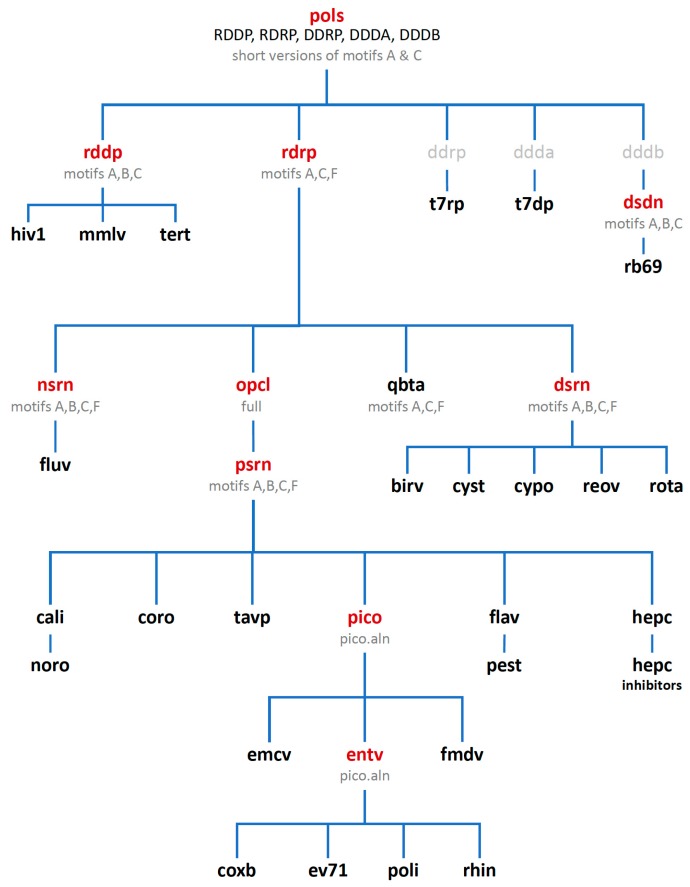
The alignment tree with polymerase sets used for superposing structures. Set names shown in red contain partial or representative family structures, those shown in black contain all solved structures relevant to that particular set (except **hiv1** and **hepc**, see text), and sets shown in grey have not yet been made as they would contain only a single viral polymerase at this point.

**Figure 4 viruses-11-00745-f004:**
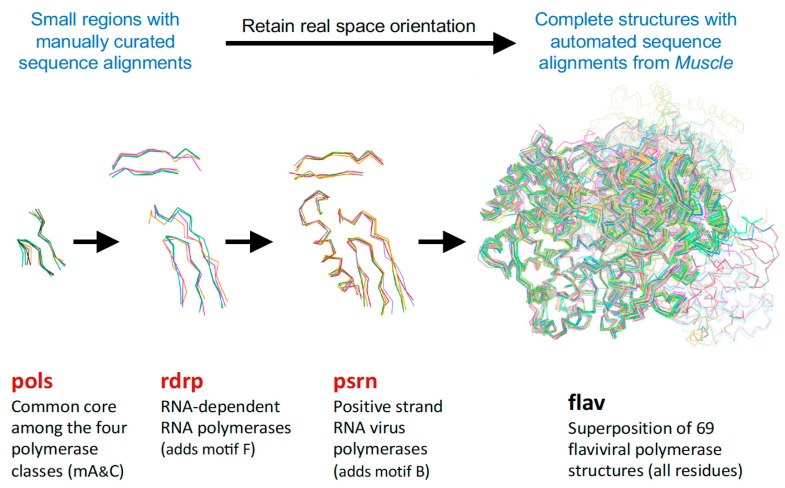
Illustration of how the size of the superposed region increases with structural similarity. The **pols** set has five structures, and the RdRP from that set is used to anchor the **rdrp** set composed of positive-, negative-, and double-stranded RNA virus polymerases. The PSRN structure from there in turn anchors the **psrn** set with more positive-strand polymerases, and the FLAV member of that set then anchors the **flav** set. Note how higher similarity allows the superposition region to increase, e.g., adding motif F for **rdrp**, then motif B for **psrn**, and entire polymerase domain for **flav**.

**Figure 5 viruses-11-00745-f005:**
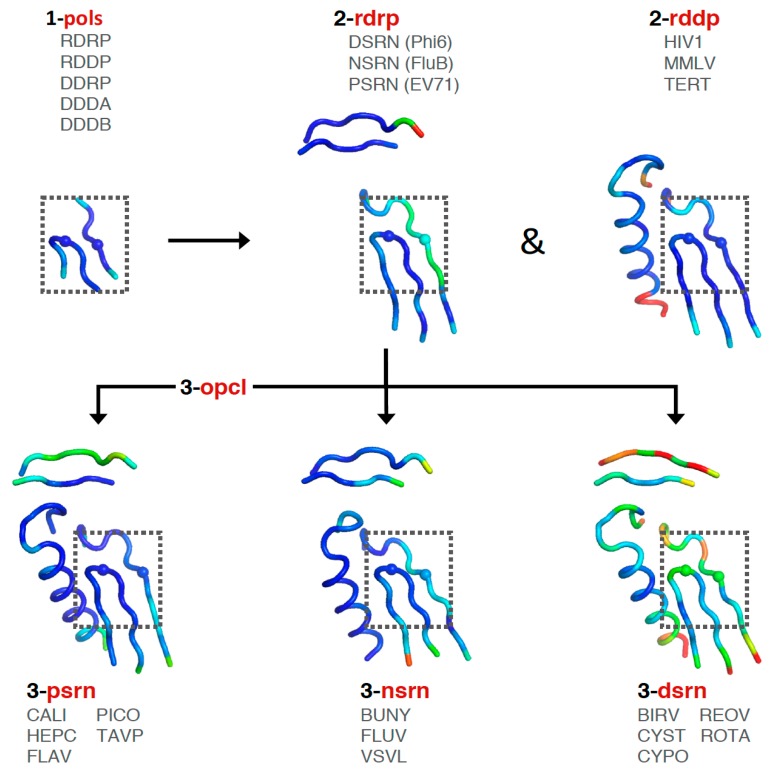
Growth of the alignment tree shown by the average structures of the aligned Cα atoms that have been colored by variance in Cα positions to show structural heterogeneity. The darker blue color represents higher similarity within each set, but note the color ranges are autoscaled differently in the different images. The box outlines the universal high-similarity region of motifs A and C used in the **pols** set and the Cα atoms of the active site Asp residues are shown as spheres.

**Figure 6 viruses-11-00745-f006:**
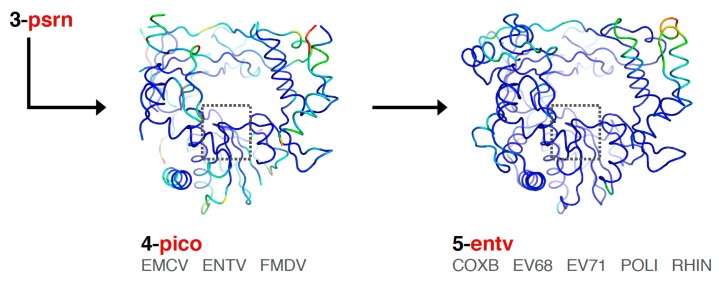
Expansion of the **psrn** set to complete structures for the **pico** and subsequent **entv** sets. Coloring shows Cα position variances and the box encompasses the **pols** set region, as in Figure 5.

**Figure 7 viruses-11-00745-f007:**
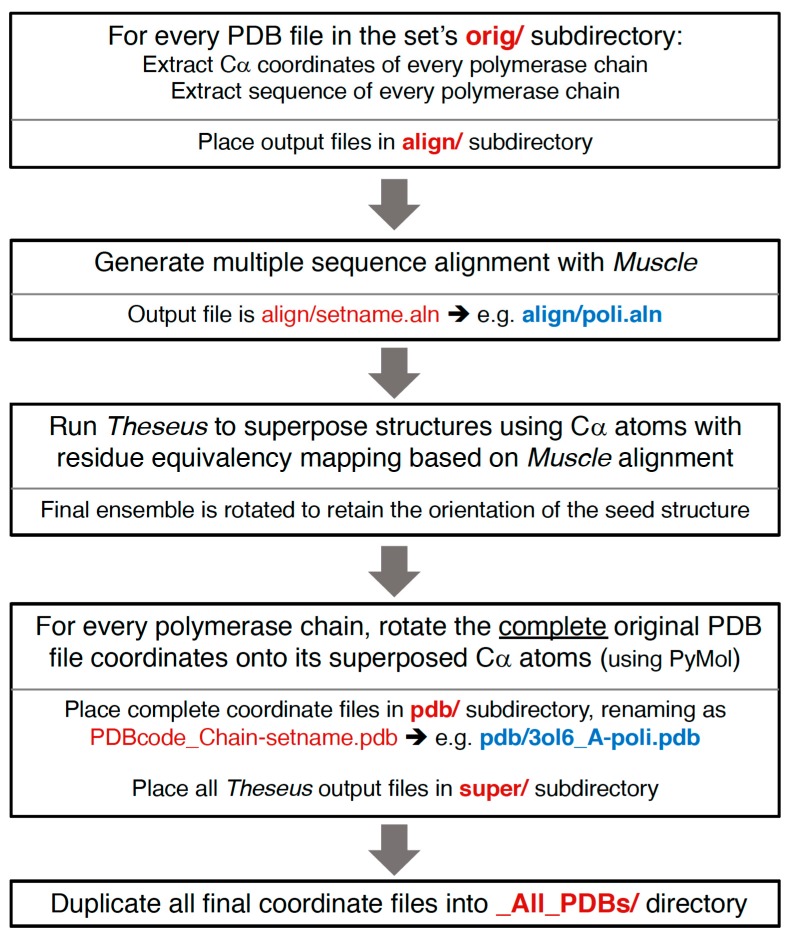
Structure superpositioning procedure that is carried out for every **set** in the alignment tree.

**Figure 8 viruses-11-00745-f008:**
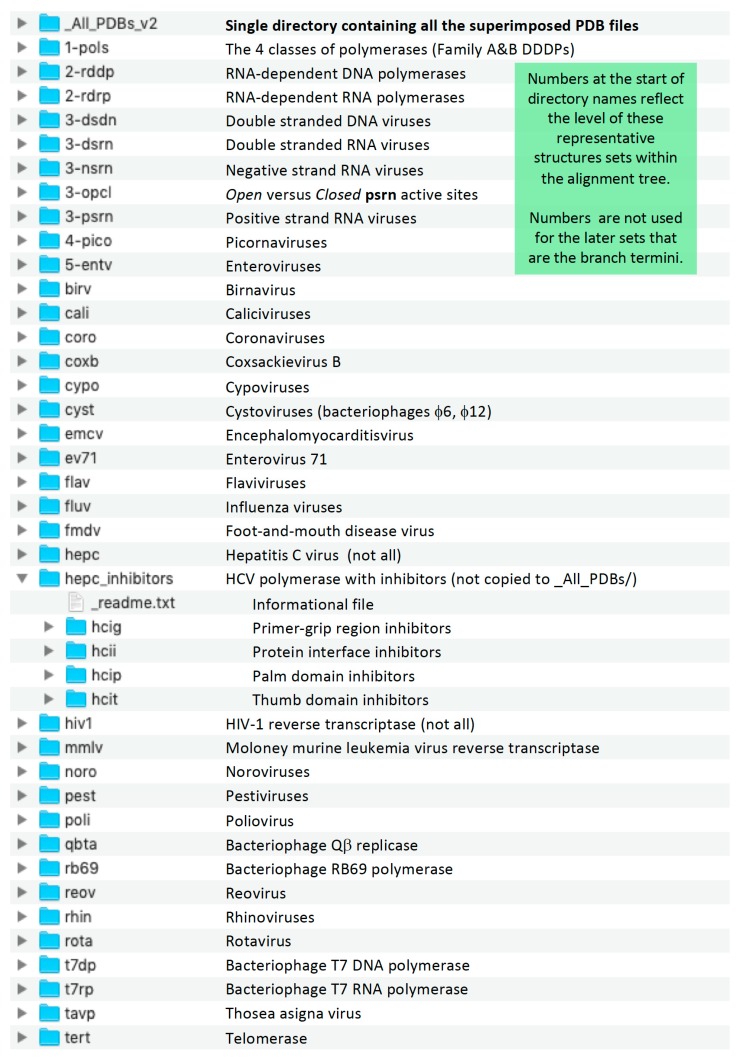
Listing of all the directories for the **sets** in the polymerase superposition tree. The names of the early stage sets where substructures are used are preceded by numbers to sort then at the top of the list (also red in Figure 2), and the Hepatitis C virus inhibitor sets are in their own subdirectory. The final output PDB files from all sets are duplicated into the _All_PDBs directory for convenience.

**Figure 9 viruses-11-00745-f009:**
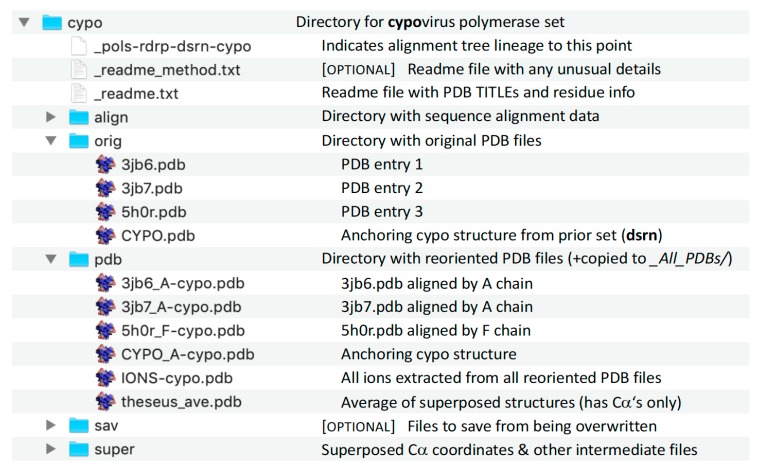
Content of the individual superposition set directories demonstrated with the **cypo** set.

**Figure 10 viruses-11-00745-f010:**
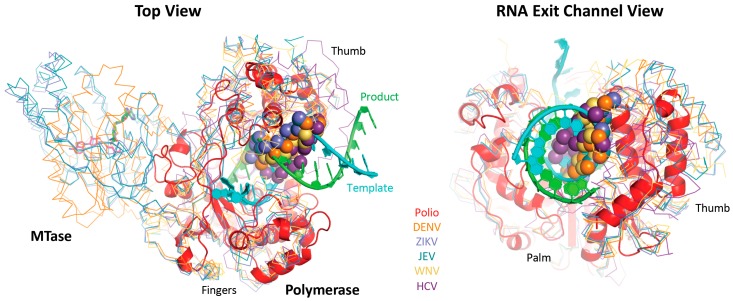
Comparison of poliovirus 3D^pol^ elongation complex with flavirial structures showing how the priming loop (spheres) from the thumb domain lies in the RNA exit channel observed with 3D^pol^. This prevents in vitro binding and initiation on a double stranded RNA helix substrate and promotes de novo initiation with a single stranded RNA template where the nascent product is believed to somehow displace this priming loop as it exits the polymerase domain. NS5 has an additional methyl transferase domain that is not included in the structure alignments, and this domain is observed in different orientations in dengue versus Japanese encephalitis and Zika virus structures, all of which copurify with bound S-adenosyl-homocysteine (sticks). The figure is drawn with coordinate files 3ol7_A-poli.pdb (polio), 5jjr_A-flav.pdb (DENV), 5tmh_A-flav.pdb (ZIKA), 4k6m_A-flav.pdb (JEV), 2hfz_A-flav.pdb (WNV), and 2xi2_A-hepc (HCV).

**Figure 11 viruses-11-00745-f011:**
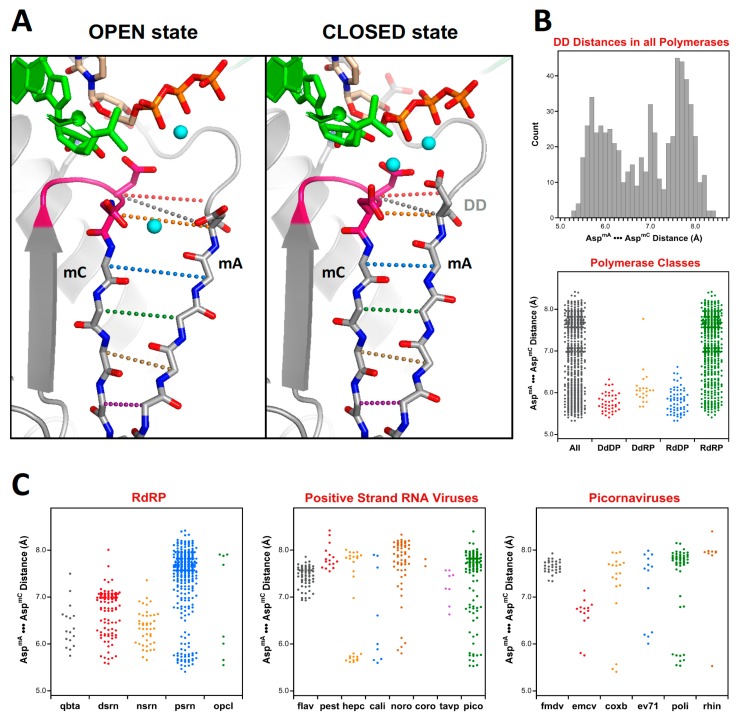
Analysis of active site closure mechanism in viral polymerases. (**A**) Comparison of the open and closed states of polio polymerase, where the closed state features a fully hydrogen-bonded anti-parallel β-sheet between motifs A and C, but in the open state the sheet is frayed near the active site. The distance between the two active site aspartates (DD, shown in grey dots) provides a convenient measure of the active site conformation. (**B**) Histogram of such DD distances for all the polymerases in the superposition set. The bottom panel is a dot plot of the histogram data (All) alongside dot plots that further categorize the distances by polymerase type. (**C**) A further breakdown of the DD distances observed in all RdRPs, then in the positive strand RNA virus polymerases, and finally in six families of picornaviruses.

**Figure 12 viruses-11-00745-f012:**
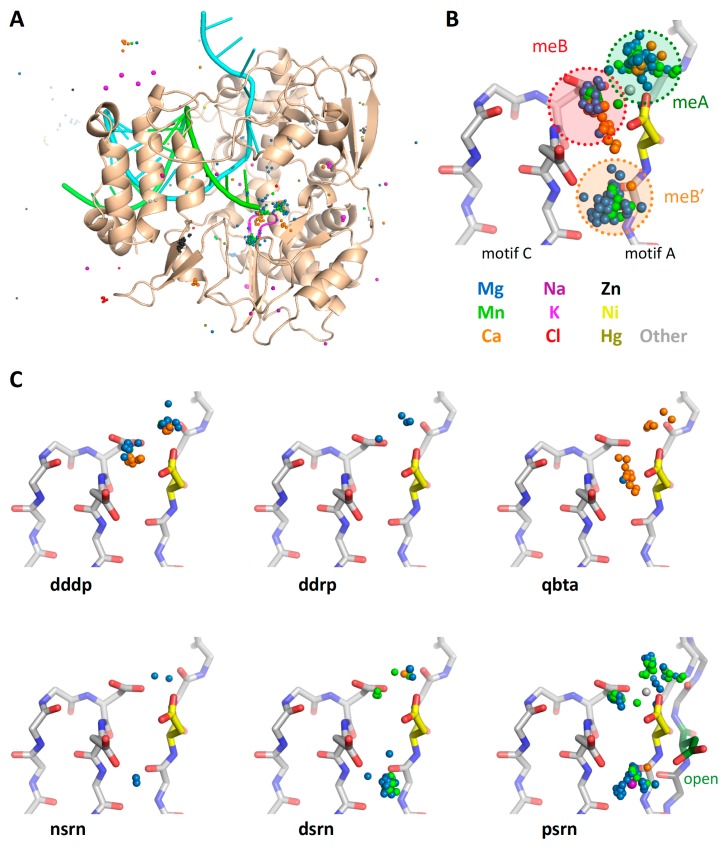
Presence of a third metal binding site in the viral RdRPs. (**A**) The locations of all metal ions from all the structures in the _All_PDBs/ directory, superimposed on the poliovirus polymerase structure as viewed from the NTP entry channel for a visual reference. (**B**) Detailed view of the active site showing all nearby ions in the indicated colors together with the poliovirus polymerase closed active site structure for reference. The clusters of ions reflect the catalytic metal A that arrives as a complex with an NTP and metal B that is prebound to the polymerase, and a third site labeled meB’. (**C**) Further breakdown of the bound metals according to polymerase type; the DdDP and DdRP structures show a mixture of magnesium and calcium in the meA and meB sites, while structures of the RdRP from bacteriophage Qbeta show mostly calcium ions. The ions in the meB’ site are mostly from viral RdRP structures, where it has been observed in negative-, double-, and positive-strand RNA virus polymerases. The final panel shows both closed (yellow Asp) and open (green Asp) states of the poliovirus active site to demonstrate how the motif A aspartate reorients from the meB’ site to the meB site during active site closure, and this may facilitate the movement of the bound ion.

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
