# Peer review of "A Comprehensive Superposition of Viral Polymerase Structures"

_viruses, 2019, doi:10.3390/v11080745_

Round 1

Reviewer 1 Report

This is a well written manuscript that describes a comprehensive structure- based  superposition of  every viral polymerase structure solved, using an alignment-tree where the alignment regions grown when the similarity among the polymerases increase. The polymerases play a central role in the replication of all viruses, and the core of the active site is highly conserved. However, the rest of the elements confer divergence among structures. Sincerely I believe that this alignment will be very useful to compare the different structures and analyze with clarity the differences observed in the new structures solved.

I find no technical problems with the paper and believe the work is suitable for Viruses. However, I have some comments:

                1.- On the first page (line 38), authors says that  they described the superimposition of 643 structures from 412 different PDB entries releases up until June 30, 2020. I suppose that is 2019.

                2.- On the Figure 1(b), the loop of motif C and the end of the β-sheet of the motif A is colored in pink. Why? It is not described in the figure caption.

                3.- On figure 4, 5 and 6 they colored the structures by variance in  Cα positions. It would be useful to add a bar that show the color related with the variance.

                4.- There are some structures that contains structural changes that affect the active site or the elements that are used for the first orientation of the polymerase  (for example, the polymerase of EV71 have conformationals changes in the loop located on the motif B (PDB id 3N6L, 3N6M and 3L6N), or one structure of EMCV polymerase presents a comformational change that displace the Asp of the motif A 10 Å (4NZ0) from their expected position. How this changes can affect the first steps in this alignment?       

Author Response

I thank the reviewer for their support of this work, and am happy to address the minor concerns raised:

The date has been corrected (line 40)
The pink coloring on motif C is for the YGDD motif at the top of motif C that serves as a visual landmark for the active site, and in work from my group we have consistently colored it in magenta. This was initially not noted in the figure legend, but that has now been corrected (line 90).
The color ranges are unfortunately subject to autoscaling by Pymol depending on the range of variances within a given superposition set and are therefore not directly comparable across different image panels. The manuscript has been updated to reflect this information, and the readers are also pointed toward a file in the super/ directory of each set that contains the raw variance information if needed (lines 223 and 423).
The reviewer is correct that significant structural deviations in the representative structures used to obtain the first orientations for each set could indeed make those orientations less than ideal, and this would affect the entire set.  For this reason, care was taken to make sure the initial representative structures have normal conformations that are free of any obvious deviations that would throw off the whole alignment.  The manuscript has been updated with a statement to that effect (line 204).
Note, however, that such deviations are generally not a problem in the context of superpositioning large bits of structure, e.g. the complete polymerase sets found at the ends of the alignment tree branches, as explained in the manuscript.

Reviewer 2 Report

The manuscript by Olve B. Peersen describes the difficulty globally comparing the large family of viral polymerases due to the poor conservation in their primary sequences. The author proposes a methodology based on a superposition on a restricted subset of catalytic residues belonging to conserved motifs A and C to provide to an overall orientation of all polymerases. Using this global orientation, structure alignment is then extended to a larger selection of residues using maximum likelihood superposition for the study of a more restricted set of viral polymerases. Such a database of pre-oriented viral polymerase structures allows more robust statistical analyses of specific structural features.

This is a well written manuscript, aside from a few typographical errors. However, I would have expected the review to be better referenced, especially concerning the few examples of analysis that are provided in part 5.

The structural overview is nicely described but it should be mentioned that in spite of the common palm/fingers/thumb domain organization, viral polymerases can display a large variety of auxiliary domains that somewhat hinder the structural superposition of polymerases from different family. Also, it is worth mentioning that in spite of a common fold, primary sequences of the polymerases can be poorly conserved even in the same family.

The methodology used for the superposition of structures using maximum likelihood based multiple alignment and an alignment tree of polymerases is clearly described and detailed, the strategy is simple enough to be apply to other protein families that display similar attribute - high sequence heterogeneity but roughly similar global organization.

The author also provides a few nice examples of how their global superposition of polymerase structures can be used to highlight some features specific to a subset of polymerase. However, few of the examples given do not seem to benefit directly from the analyses of the large superposition of structures but only highlight features that are already known. For example, the first example “highlighting” the role of the “priming loop” in the NS5 polymerase of Zika virus was already discussed in the original paper from Zhao et al. (2017), such characteristics are also highlighted in other de-novo polymerases (Butcher et al., 2001; Laurila et al., 2002, Tao et al., 2002; Lu et al., 2008; Choi and Rossmann, 2009; Appleby et al., 2015.

In the last example given, in the analysis of metals bound at the metal binding site, the article would benefit of a more rigorous statistical analysis. 

Specific Comment :

Line 26-27 :  a more specific example for  “tiny bacteriophage” could be used , since bacteriophage represents a broad family of viruses with a large difference of size  (from 20 to 200 nm)

Line 34 :  Replace “is great detail” by “in great detail”

Line 38 : “June 30, 2020” should be correct to “June 30, 2019”

Line 52 : the hyperlink is not correctly written and should be replaced by www.theseus3d.rog

Line 355-356 : Reference to  Zhao et al. Nature Communications volume 8,  14762 (2017), which describe in details the role of the priming loop in the flaviviral NS5 RdRP could be added.

Figure 1 (b) : Part of the palm domain is colored in pink (the loop connecting the two beta strands) but it is not discussed in the legend or the main text.

In Figure 9 : The left beta strand change of color between A (gray), B (green) and C (yellow) but the difference is not clearly stated. I guess it reflects the "open" and "close" state but his need to be confirmed ?

Author Response

I would like to thank the reviewer for their comments and providing valuable guidance for the revision of the manuscript.  I hope the new version will adequately address the reviewer concerns, and 

I do understand the reviewer's point that the article could be better referenced, in particular within section 5 where I provide some example analyses of the superposed structures. The issue, in my mind, it that because these analyses are based on comparing a large number of structures and all of these contribute the observations and conclusions, it become difficult to decide which structures to reference.  I therefore made the conscious decision to focus this manuscript on the methodology for the alignment and only make indirect references to specific structures through their PDB entry codes when pertinent. I also made the decision to not explicitly reference all 414 PDB entries in the dataset as doing so seemed excessive and redundant with the PDB database.

Regarding the specific inclusion of only the Zika virus NS5 structure when discussing the priming loop, I changed Figure 10 to better show that this interaction is universal among flaviviruses by including structures of all solved flaviviral polymerases (DENV, ZIKA,  JEV, WNV, and HCV) and listed all the PDB entries used in the legend.

With respect to the metal sites, I did not carry out a formal statistical analyses of the observed metals ions and their binding site, but I did significantly expand the figure and added a supplemental figure to better show the bound ions.  The new Figure 12A shows all polymerase-associated ions superposed on the entire poliovirus polymerase structure as a positional reference, while Figure 12B zooms in and shows a more detailed view of the metals at the active site itself.  Next, Figure 12C breaks these bound ions structures down further by showing the ions associated with different classes of polymerases and viruses, much the way the Asp-Asp distances in Figure 11 were categorized. Finally, the ions have been color coded by type, allowing the reader to quickly identify bound Mg, Mn, Ca, .... ions.

Corrections to the more specific comments are as follows:

Line 26-27:  "tiny" has been removed and the sentence restructured (line 26).

Line 34:  typographical error has been fixed (line 34)

Line 38: date has been corrected (line 40)

Line 52:  Name and hyperlink for thesesus3d.org has been corrected (line 171)

Figure 1B: Legend has been updated to mention the pink in the palm domain is the YGDD motif at the active site (line 90)

Figure 9 (now 12): Figure has been revised, and the coloring of yellow-vs-green for closed-vs-open active site is now described in the legend (line 608).